# Impaired Endothelial Function in Convalescent Phase of COVID-19: A 3 Month Follow Up Observational Prospective Study

**DOI:** 10.3390/jcm11071774

**Published:** 2022-03-23

**Authors:** Luca Santoro, Lorenzo Falsetti, Vincenzo Zaccone, Antonio Nesci, Matteo Tosato, Bianca Giupponi, Maria Cristina Savastano, Gianluca Moroncini, Antonio Gasbarrini, Francesco Landi, Angelo Santoliquido

**Affiliations:** 1Department of Cardiovascular and Thoracic Sciences, Fondazione Policlinico Universitario Agostino Gemelli IRCCS, 00168 Rome, Italy; luca.santoro@policlinicogemelli.it (L.S.); antonio.nesci@policlinicogemelli.it (A.N.); angelo.santoliquido@policlinicogemelli.it (A.S.); 2Department of Emergency Medicine, Internal and Sub-Intensive Medicine, Azienda Ospedaliero-Universitaria “Ospedali Riuniti”, 60166 Ancona, Italy; drfalsetti@yahoo.it; 3Geriatrics Department, Fondazione Policlinico Universitario Agostino Gemelli IRCCS, 00168 Rome, Italy; matteo.tosato@policlinicogemelli.it (M.T.); francesco.landi@policlinicogemelli.it (F.L.); 4Department of Emergency Medicine, Fondazione Policlinico Universitario Agostino Gemelli IRCCS, 00168 Rome, Italy; bianca.giupponi@policlinicogemelli.it; 5Department of Ophthalmology, Fondazione Policlinico Universitario Agostino Gemelli IRCCS, 00168 Rome, Italy; mariacristina.savastano@gmail.com; 6Department of Experimental and Clinical Medicine, Clinica Medica, Azienda Ospedaliero-Universitaria “Ospedali Riuniti”, 60166 Ancona, Italy; g.moroncini@univpm.it; 7Department of Medical and Surgical Sciences, Fondazione Policlinico Universitario Agostino Gemelli IRCCS, 00168 Rome, Italy; antonio.gasbarrini@policlinicogemelli.it; 8Università Cattolica del Sacro Cuore, 00168 Rome, Italy

**Keywords:** endothelial dysfunction, flow-mediated dilation, COVID-19, post-acute COVID-19 syndrome

## Abstract

Background: Endothelial dysfunction has a role in acute COVID-19, contributing to systemic inflammatory syndrome, acute respiratory distress syndrome, and vascular events. Evidence regarding COVID-19 middle- and long-term consequences on endothelium are still lacking. Our study aimed to evaluate if COVID-19 severity could significantly affect the endothelial function after three months from the acute phase. Methods: We assessed endothelial function in outpatients with previous COVID-19 three months after negative SARS-CoV-2 molecular test by measuring flow-mediated dilation (FMD) in patients categorized according to a four-variable COVID-19 severity scale (“home care”; “hospital, no oxygen”; “hospital, oxygen”; “hospital requiring high-flow nasal canula, non-invasive ventilation, invasive mechanical ventilation, or extracorporeal membrane oxygenation”). FMD difference among COVID-19 severity categories was assessed with analysis of variance; we further clarified the relationship between FMD and previous COVID-19 severity with multivariate logistic models. Results: Among 658 consecutive COVID-19 subjects, we observed a significant linear trend of FMD reduction with the increase of the COVID-19 category (*p* < 0.0001). The presence of endothelial dysfunction was more frequent among hospitalized patients (78.3%) with respect to home-care patients (21.7%; *p* < 0.0001). COVID-19 severity was associated with increased endothelial dysfunction risk (OR: 1.354; 95% CI: 1.06–1.71; *p* = 0.011) at multivariate binary logistic analysis. FMD showed a significant direct correlation with PaO2 (*p* = 0.004), P/F ratio (*p* = 0.004), FEV1 (*p* = 0.008), and 6MWT (*p* = 0.0001). Conclusions: Hospitalized COVID-19 subjects showed an impaired endothelial function three months after the acute phase that correlated with pulmonary function impairment. Further studies are needed to evaluate if these subjects are at higher risk of developing pulmonary disease or future cardiovascular events.

## 1. Introduction

Knowledge about consequences on middle- and long- term health status in the infection by severe acute respiratory syndrome coronavirus 2 (SARS-CoV-2), the causative agent of the novel coronavirus disease (COVID-19), is still limited [1].

It is widely known that SARS-CoV-2 is able to infect not only the lungs, but also several other tissues and organs [2]. In particular, SARS-CoV-2 can cause vascular damage both by direct endothelial infection—mediated by the binding to the ACE2 receptor of the endothelial cell—and indirect endothelial activation due to abnormally raised systemic inflammation [3,4,5]. For these reasons, the endothelium could be considered both as a target organ and as an effector of inflammation and thrombosis, playing a critical role in COVID-19 development [6]. Accordingly, clinical evidence in the acute phase of disease shows an increased incidence of vascular events, such as thromboembolism, stroke, and acute myocardial infarction, that can be caused by both macrovascular and microvascular alterations [7,8,9,10,11,12].

After SARS-CoV-2 RNA nasopharyngeal swab negativization, in the convalescent phase of the disease, some patients describe the persistence of a heterogeneous group of signs and symptoms for more than 12 weeks, not justified by an alternative diagnosis, identified as “post-acute COVID-19 syndrome” [13,14,15]. This syndrome is actually not well-defined, requiring further diagnostic and therapeutic improvements. 

To date, the middle- and long-term consequences of COVID-19 on vascular function and its role on the development of the post-acute COVID-19 syndrome are not completely known [15,16]. Some studies have already shown persistently damaged endothelium [17] and impaired endothelial dilatation in post-COVID-19 patients [18,19]; however, data regarding the role of the severity of acute disease on a following endothelial dysfunction are lacking [20].

The aim of this study was to evaluate, in a large prospective cohort of patients, if COVID-19 severity could have effects on endothelial function, measured by flow-mediated dilation (FMD), in the convalescent phase of disease. 

## 2. Materials and Methods

### 2.1. Study Setting and Design

We conducted an observational prospective study based on the “Post-acute COVID-19 Day Hospital Unit registry—Fondazione Policlinico Universitario Agostino Gemelli IRCCS in Rome, Italy”, a prospective monocentric observational registry including outpatients with previous COVID-19 infection referred for a post-acute COVID-19 recovery health check with a complete clinical and multidisciplinary assessment. Exclusion criteria were: (i) age less than 18 years, (ii) patients lost at the follow-up. Subjects who did not survive to the acute phase of COVID-19 were not considered in the analyzed cohort. The study was performed in accordance with the Declaration of Helsinki and was approved by the ethics committee of the Catholic University of Rome (protocol ID number: 003220/20). Written informed consent was obtained from all participants before entry into the study.

### 2.2. Patients

A total of 658 consecutive patients with previously confirmed SARS-CoV-2 infection and COVID-19 and aged ≥18 years old were referred for a post-acute COVID-19 recovery health check 3 months from the negative SARS-CoV-2 molecular test to a post-acute care service, established in 21 April 2020, at the “Fondazione Policlinico Universitario Agostino Gemelli IRCCS” in Rome, Italy. The analyzed cohort refers to the timeframe between 21 April 2020 and 31 March 2021, comprising unvaccinated patients infected by the original strain. Patients were offered a comprehensive and interdisciplinary medical assessment, which has already been detailed elsewhere [21]. Specifically, data regarding demographic, clinical, biochemical, instrumental, and COVID-19 characteristics were collected in an electronic database and managed using REDCap electronic data capture tools in order to minimize missing inputs and allow real-time data validation and quality control.

COVID-19 disease severity was characterized according to the current definition [22,23]; however, in order to harmonize subgroup number, we adopted the following severity scale, defined by the highest of these four categorical variables:The “Home care” group, obtained by merging the following two original categories: (a) patients not admitted to hospital with resumption of normal activities, and (b) patients not admitted to hospital, but unable to resume normal activities;The “Hospital, no oxygen” group, including patients admitted to hospital but not requiring supplemental oxygen;The “Hospital, oxygen” group, including patients admitted to hospital, requiring supplemental oxygen, but not requiring high-flow nasal canula (HFNC), non-invasive ventilation (NIV), invasive mechanical ventilation (IMV), or extracorporeal membrane oxygenation (ECMO);The “Hospital, NIV or ICU” group, obtained by merging these two original categories: (a) patients admitted to hospital requiring HFNC or NIV, and (b) patients admitted to hospital requiring IMV, ECMO, or both.

### 2.3. Vascular Assessment

The endothelial function was assessed by measuring the brachial artery reactivity, since this method represents the most well-established technique in adults. This evaluation leads to the assessment of FMD, which represents the nitric oxide-mediated vasodilatation produced by increased flow after a period of ischemia or endothelium-dependent vasodilatation. FMD evaluation was performed according to current guidelines [24,25]: briefly, after a 15–20 min supine rest, the right brachial artery was scanned over a longitudinal section of 5–7 cm over the antecubital fossa; its diameter was evaluated from the intima of both anterior and posterior walls; then, a pulse Doppler velocity signal was recorded. After this basal measurement, a blood pressure cuff around the forearm distal to the target area was inflated to a pressure of 250 mmHg for 5 min and then abruptly deflated, after which a second scan was performed continuously for 90 s, to measure changes in diameter after reactive hyperemia. A pulse Doppler velocity signal is also obtained no more than 15 s after deflation to measure the maximal hyperemic velocity. FMD data were expressed as percentage increases relative to baseline diameters.

All ultrasound scans were performed by a single skilled sonographer blinded to the subject’s clinical characteristics, in a quiet, temperature-controlled room, in the morning (8.00–10.00 a.m.), to avoid the reported circadian variation in endothelial function [26]. All subjects refrained from exercise and from ingesting food and any vasoactive substances (i.e., tobacco and coffee) for at least 12 h before the examination. All the evaluations were performed while breathing room air. Intra-rater reliability was assessed using intra-class correlation coefficient (ICC).

### 2.4. Statistical Analysis

Age, body mass index (BMI), FMD, and C-reactive protein (CRP) were collected as continuous variables. Sex, arterial hypertension, and diabetes mellitus were collected as binary variables. BMI was recoded as a categorical variable according to WHO BMI categories [27]. Age was categorized in three classes: 18–64, 65–74, and more than 75 years old. COVID-19 severity and smoking status were synthesized as categorical variables: COVID-19 severity was synthesized as “home care”; “hospital, no oxygen”; “hospital, oxygen”; and “hospital, NIV, or ICU”. Smoking status was categorized as “never smoker”, “ex-smoker”, and “active smoker”. We also dichotomized COVID-19 severity into “home care” and “hospital care”, in which we merged “hospital, no oxygen”; “hospital, oxygen”; and “hospital, NIV or ICU”. 

We analyzed the association between FMD and a previous hospitalization for COVID-19, adopting the above-mentioned dichotomous variable and ROC curve analysis. According to the results of this analysis, we then prepared a dichotomous variable of FMD, adopting the cutoff of ≤7.10% to identify a significant endothelial dysfunction. 

Continuous variables were tested for normality with the Kolomogorov–Smirnov test. Normally distributed variables were presented as mean and standard deviation and compared with *t*-test or analysis of variance (ANOVA). The relationship between continuous variables was tested first with linear regression. Non-normally distributed variables were presented as median and interquartile range and compared with the Mann–Whitney U test or Kruskal–Wallis H test. Multiple comparisons were assessed with both the least squared method and with the Bonferroni correction. Categorical and dichotomous variables were presented as number and percentage and compared with the chi-squared test. We also tested the trend of dependent continuous variables among categorical variables adopting ANOVA with polynomial linear contrast. The relationship between FMD and disease severity was further explored with (i) multinomial logistic regression model considering disease severity as the dependent variable and binary FMD as the independent variable, and (ii) a binary logistic regression model considering FMD as the dependent variable and disease severity as the independent variable, corrected by age, sex, BMI, arterial hypertension, diabetes mellitus, and CRP levels. We analyzed the relationship between the continuous FMD variables and respiratory variables (PaO2, P/F, FEV_1_, and 6MWT) with multiple regression models, maintaining the constant value to reduce the bias and choosing best-fitting trendline according to the best r^2^ index. We considered as significant all the comparisons at the level of *p* < 0.05. Statistical analysis was performed with SPSS 13.0 for Windows Systems.

## 3. Results

### 3.1. Baseline Characteristics of the Sample

From the original cohort of 685 patients, we enrolled 529 subjects who had a FMD test performed at the control visit. The subpopulation of subjects excluded for missing FMD at the control visit (*n* = 156) did not significantly differ from the population considered in age, BMI, vascular risk factors, and COVID-19 severity. We categorized the patients in four different COVID-19 severity categories using a modified version of the currently adopted categorization [22,23]. Baseline characteristics of the considered patients are summarized in Table 1. Regarding FMD measurements, we observed a high intra-class correlation coefficient (ICC = 0.926; *p* = 0.029).

### 3.2. Relationship between FMD and COVID-19 Severity

We analyzed the differences of the mean FMD among COVID-19 severity categories with ANOVA, as shown in Table 2 and Table 3. When evaluating the mean of FMD among COVID-19 severity categories, we observed a significant linear FMD reduction with the increase of the COVID-19 severity category (*p* = 0.0001), as shown in Table 2 and Figure 1. Multiple comparisons between COVID-19 categories from analysis of variance showed a statistically significant difference in mean FMD between the “home care” group and the other three groups of hospitalized patients (“hospital, no oxygen”; “hospital, oxygen”; and “hospital, NIV, or ICU”), as shown in Table 3. Moreover, when comparing the mean FMD between the “home care” and the “hospital care” group, we observed that the first group had a significantly higher mean FMD (12.03 ± 4.34%) than the second group (10.20 ± 4.54%; *p* = 0.0001).

FMD was significantly associated to a previous hospitalization for COVID-19 at ROC curve analysis (AUC: 0.624; 95% CI: 0.581–0.665; *p* < 0.0001), with the best cut-off point at 7.14% (sensitivity: 29.28; 95% CI: 24.4–34.6; specificity: 87.02; 95% CI: 81.7–91.3; positive likelihood ratio: 2.26; 95% CI: 1.5–3.3; negative likelihood ratio: 0.81; 95% CI: 0.7–0.9). Thus, to identify the presence of a severe endothelial dysfunction, we adopted a dichotomous FMD with a cutoff of 7.10%, according to the analysis of our data and as recently reported by other authors [28]. When considering the dichotomous FMD variable and the COVID-19 disease categories, we observed a significant difference (*p* = 0.0001, chi-squared test) in the distribution according to FMD as shown in Figure 2 and Table 4. In particular, when considering the group of subjects with a normal FMD (>7.10%), we observed that 44.5% was among the “home care” patients; on the other hand, when considering a pathologic FMD (≤7.10%), 78.3% of these subjects had a previous hospitalization for COVID-19, with the greatest prevalence in the group of patients hospitalized with oxygen (39.2%).

The multinomial regression analysis showed that the presence of a FMD ≤ 7.10% increased about three times the risk of belonging to the category of “hospital care” patients than “home care” patients (OR: 2.899; 95% CI: 1.801–4.666; *p* = 0.0001); similarly, each category of “hospital care” patient is associated to an increased risk of a severe endothelial dysfunction, as shown in Table 5. The binary logistic regression analysis showed that an increase in the disease severity category was significantly associated to an increased probability of a subsequent pathologic FMD, with an odds ratio of 1.354 (95% CI: 1.06–1.71; *p* = 0.007), as shown in Table 6. Significantly, according to this model’s results, this effect was independent to CRP, arterial hypertension, diabetes mellitus, smoke status, body mass index, and sex, while increasing age could have concurred to the observed effect.

### 3.3. Relationship between FMD and Pulmonary Function Outcomes

We observed a significant association of arterial partial pressure of oxygen (PaO2), ratio of PaO2 to fractional inspired oxygen (P/F), forced expiratory volume in one second (FEV1), and six minutes walking test (6MWT) with FMD, as shown in Figure 3. Of note, FMD had a cubic relationship with PaO2 (Figure 3A; r^2^ = 0.030; *p* = 0.004), P/F (Figure 3B; r^2^ = 0.030; *p* = 0.004), FEV1 (Figure 3C; r^2^ = 0.026; *p* = 0.008), and 6MWT (Figure 3D; r^2^ = 0.066; *p* = 0.0001). The low r^2^ values with a high significance of the association are suggestive of the fact that other factors than FMD can affect the variability of these values (as, for example, a previous COPD in the case of FEV1) but that this association is significant, meaning that FMD has a clear association with these variables.

## 4. Discussion

In this observational prospective study, we found that subjects with previous hospitalized COVID-19 showed an impaired endothelial function compared to home care COVID-19, after three months from the acute phase, and the degree of this dysfunction was related to COVID-19 severity. Moreover, in this setting, endothelial dysfunction was found to be directly related with variables of pulmonary function that could play a critical role in the development of post-acute COVID-19 syndrome.

Evidence suggests that a dysregulation of the endothelial function, induced by SARS-CoV-2 infection, could have a central role in the COVID-19 pathogenesis [2]. In fact, the vascular endothelium has a critical role in cytokine dysregulation observed in the acute respiratory distress syndrome [5,29]. Moreover, the pro-thrombotic phenotype and the diffuse intravascular coagulation described during COVID-19 could be expression of an endothelial dysfunction [30]. Thus, it is possible to speculate that the dysregulation of endothelial function could represent one of the most important steps in the evolution of COVID-19 from a local infection to a severe inflammatory disease with lung and systemic involvement [31]. On the other side, epidemiological data suggest that patients with an endothelial dysfunction pre-existing before SARS-CoV-2 infection could be more at risk of developing severe forms of COVID-19 [32,33,34].

Our knowledge of the effects of endothelial damage after resolution of the acute phase of COVID-19, particularly on the post-acute COVID-19 syndrome, is limited [18,19], and there are no study about the role of disease severity on endothelial dysfunction [20]. However, a position statement of experts’ opinions emphasized the urgent need of further defining the endothelial function in subjects with a previous COVID-19 with functional tests, such as FMD [15].

Our observational prospective study evaluated the data from a large registry performed on convalescent COVID19 subjects, specifically analyzing 529 patients with different degrees of COVID-19 severity whose endothelial function was assessed with FMD after three months from the acute disease. Evaluating the mean FMD according to disease severity, we found a significant trend towards an endothelial function worsening with COVID-19 severity increase (Figure 1, Table 2), with a significant difference when comparing the mean FMD of “home care” patients with each category of hospitalized patients, as shown in Table 3. This observation was furtherly confirmed when we categorized the whole cohort into “home care” and “hospital care” patients: in this second group, we observed a significantly increased prevalence of a pathologic FMD, when compared with “home care”.

Categorizing patients to identify the presence of a severe endothelial dysfunction, according to the analysis of our data and as recently reported by other authors [28], we observed that subjects with a worse endothelial function more frequently (78.3%) had a previous hospitalization for COVID-19, with a greater prevalence of patients requiring oxygen therapy (39.2%). This prevalence was reduced among patients requiring NIV or ICU (18,3%), but this datum could have been affected by the high in-hospital death rate described for this group, which could reduce the number of observed cases at follow-up [35]. 

Moreover, we underlined that the presence of a severe endothelial dysfunction increased about three times the risk of belonging to “hospitalized” patients than “home care” patients. Interestingly, the multivariate regression analysis showed that the relationship between COVID-19 severity and endothelial dysfunction was independent from other variables that are commonly considered as risk factors for endothelial dysfunction (CRP, BMI, arterial hypertension, cigarette smoking, diabetes), resulting in being dependent only in terms of age. 

These last results are very interesting: clinical and epidemiological studies indicate that subjects characterized by older age (>65 years), obesity, arterial hypertension, diabetes, and coronary artery disease have a higher risk of developing a moderate–severe form of COVID-19 [32,33]. Hence, the presence of a markedly reduced FMD in subjects with a more severe form of the disease could merely represent a pre-existing endothelial dysfunction. The literature data indeed underline the fact that the acute inflammation caused by COVID-19 could represent another potential damage of endothelium that could lead some of these subjects to develop clinically relevant vascular events [7,8,9,10,11]. For this reason, persistence or worsening of this dysfunction could be associated to the severe disease course, predisposing to the development of a chronic disease such as post-acute COVID-19 syndrome and increasing the cardiovascular risk of these subjects.

Surprisingly, our analysis underlined the fact that while the FMD continues to be correlated to COVID-19 severity after 3 months, CRP measurement did not maintain its association with disease severity (data not shown). We can speculate that subjects developing a moderate–severe form of COVID-19 maintain a local sub-inflammatory status that could remain localized in the vessels where it is able to maintain an endothelial dysfunction in the middle term, without inducing production of systemic inflammatory markers, such as CRP. This could suggest FMD as a surrogate marker of persistent endothelial inflammation in subjects infected previously with COVID-19, which is undetectable with the ordinary laboratory measurements and could potentially identify subjects at risk of post-acute COVID-19 syndrome. 

In line with previous evidence of direct correlation between endothelial and pulmonary dysfunction in convalescent COVID-19 patients [19], we found a direct correlation between FMD and some variables of pulmonary function such as PaO2, P/F, FEV1, and 6MWT. These data could suggest a role of endothelial dysfunction in development of functional and clinical manifestations of post-acute COVID-19 syndrome. In fact, we could speculate that the principal symptoms of long COVID-19, such as fatigue, malaise, and dyspnea, could be the clinical manifestation of a sub-inflammatory state that cause both an endothelial dysfunction and an impairment of pulmonary function. 

### 4.1. Study Limitations

We wish to underline that our cohort is characterized by a balanced number of non-hospitalized (48.6%) and hospitalized (51.4%) patients, but that the single categories of hospitalized patients are smaller than the single category of non-hospitalized patients. 

The clinical evaluation of the endothelial function was carried out by calculating FMD. This technique that encloses different methodological approaches that can limit its validity and comparability without the support of dedicated software. Moreover, we did not assess FMD during COVID-19 infection but only in the convalescent phase of the disease: this is another limitation of this study. According to this limit, it is possible that patients with a greater atherosclerotic burden and a lower FMD underwent a more severe disease leading to hospitalization. Drug therapy of hospitalized patients could affect FMD; however, the wide variety of ineffective drugs adopted during the first phase of the pandemic did not allow us to assess the role of drug therapy in post-COVID FMD, especially among hospitalized patients.

### 4.2. Future Directions

Our cohort considered only unvaccinated patients infected by the original strain, while we did not consider the impact of other variants and the potential protective role of vaccines: this point needs to be clarified in cohorts designed specifically to study these aspects of the disease. Another interesting point is to assess FMD also among hospitalized patients affected by COVID-19 and non-surviving to the acute phase of the disease. Drugs adopted for in-hospital treatment of COVID-19, especially anti-IL6 and small molecules, could impact significantly on FMD function after discharge, and specific studies should be performed to address this point. 

The clinical meaning of a reduction of FMD needs to be assessed in the longer term, assessing the occurrence of cardiovascular outcomes. This analysis will require larger samples and longer observation periods, best performed in cohorts matched for cardiovascular risk factors. 

## 5. Conclusions

The results of this observational prospective study suggest the presence of a strong link between the acute phase COVID-19 severity and endothelial dysfunction at three months from the acute phase of disease, regardless of presence of other known cardiovascular risk factors; moreover, endothelial dysfunction is directly related to variables of pulmonary function. Further prospective studies are required to clarify the nature of this association and to specify the role of endothelial dysfunction in the setting of the post-acute COVID-19 syndrome and its related end-organ damage. 

## Figures and Tables

**Figure 1 jcm-11-01774-f001:**
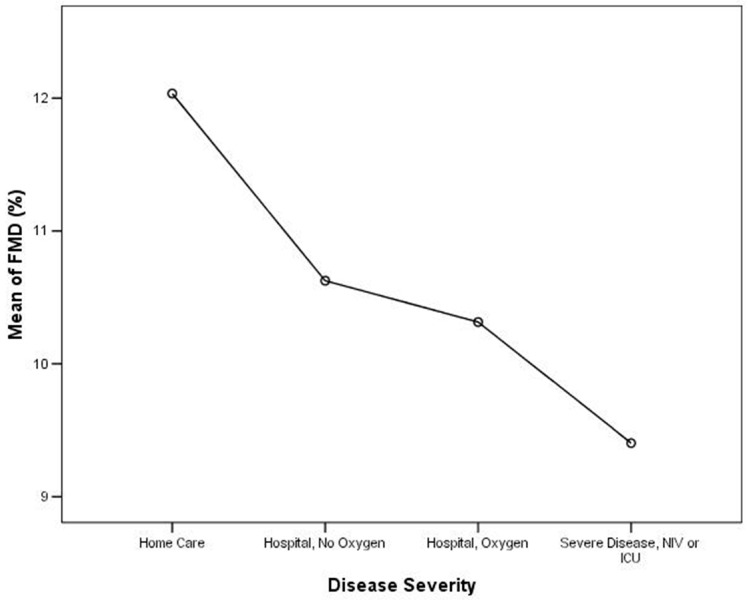
Analysis of variance for FMD and disease severity (*p* < 0.0001 for trend).

**Figure 2 jcm-11-01774-f002:**
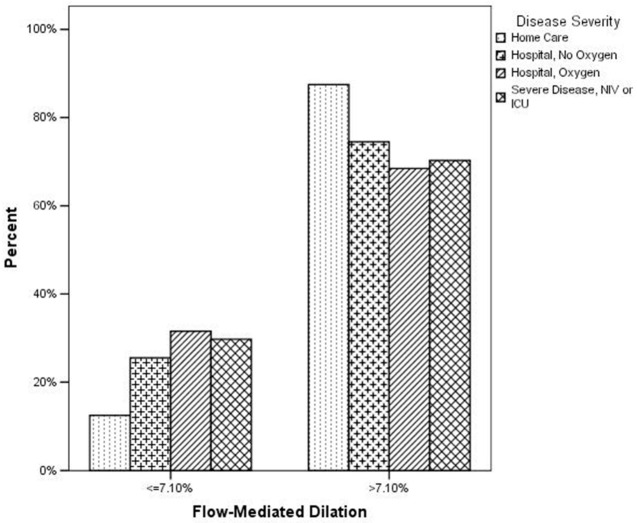
Distribution of disease severity according to flow-mediated dilation (*p* = 0.0001 at chi-squared test).

**Figure 3 jcm-11-01774-f003:**
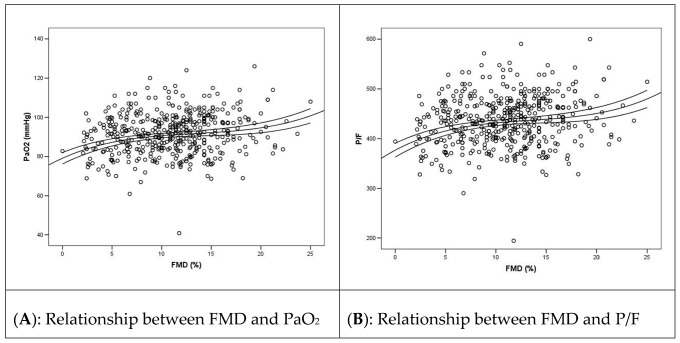
Relationship between FMD and functional outcomes in the post-COVID-19 phase.

**Table 1 jcm-11-01774-t001:** Baseline characteristics of the sample.

Variable	Cohort	Home Care	Hospital, No Oxygen	Hospital, Oxygen	Hospital, NIV/ICU	*p*
Age (mean ± SD), years	53.4 (14.9)	46.6 (14.0)	51.9 (15.0)	59.8 (13.7)	62.6 (11.4)	0.0001
Female sex (*n*, %)	335 (48.9%)	187 (27.5%)	62 (9.1%)	56 (8.2%)	30 (4.4%)	0.0001
BMI (mean ± SD), kg/m^2^	25.8 (4.28)	24.5 (4.29)	26.0 (4.79)	27.0 (4.17)	26.8 (4.06)	0.0001
COVID-19 Severity:						
Home care (*n*, %)	285 (41.6%)					
Hospital, no oxygen (*n*, %)	115 (16.8%)					
Hospital, oxygen (*n*, %)	177 (25.8%)					
Hospital, NIV/ICU (*n*, %)	108 (15.8%)					
FMD (mean ± SD), %	10.8 (4.53)	12.0 (4.34)	10.6 (4.66)	10.3 (4.55)	9.40 (4.29)	0.0001
FMD ≤ 7.10 % (*n*, %)	120 (22.7%)	26 (12.5%)	25 (25.5%)	47 (31.5%)	22 (29.7%)	0.0001
Smoking status						
Never smoker (*n*, %)	319 (46.6%)	148 (25.8%)	51 (8.9%)	65 (11.3%)	55 (9.6%)	
Active smoker (*n*, %)	50 (8.7%)	32 (5.6%)	8 (1.4%)	6 (1.0%)	4 (0.7%)	
Previous smoker (*n*, %)	204 (35.6%)	56 (9.8%)	34 (5.9%)	83 (14.5%)	31 (5.4%)	0.0001
Hypertension (*n*, %)	203 (29.6%)	49 (7.2%)	28 (4.1%)	75 (10.9%)	51 (7.4%)	0.0001
Diabetes (*n*, %)	51 (7.4%)	10 (1.5%)	4 (0.6%)	24 (3.5%)	13 (1.9%)	0.0001

Legend: BMI = body mass index; FMD = flow-mediated dilation; HF = high-flow nasal cannula; ICU = intensive care unit; IQR = interquartile range; NIV = non-invasive ventilation; SD = standard deviation.

**Table 2 jcm-11-01774-t002:** Mean of flow-mediated dilation according to different categories of disease severity (see Table 3 for multiple comparisons among COVID-19 categories).

	FMD Mean	SD	95% CI	
			Lower	Upper
Home care	12.03	4.33	11.44	12.63
Hospital, No oxygen	10.62	4.66	9.69	11.56
Hospital, oxygen	10.31	4.55	9.57	11.05
Hospital, NIV, or ICU	9.40	4.29	8.40	10.39

Legend: CI = confidence interval; FMD = flow-mediated dilation; NIV = non-invasive ventilation; SD = standard deviation.

**Table 3 jcm-11-01774-t003:** Multiple comparisons between COVID-19 categories from analysis of variance.

Severity (I)	Severity (J)	(I − J)	*p*	95% CI	
				Lower	Upper
Home care	Hospital, no oxygen	1.41	0.01	0.34	2.48
	Hospital, oxygen	1.72	0.0003	0.78	2.66
	Hospital, NIV or ICU	2.63	0.0001	1.44	3.81
Hospital, no oxygen	Home care	−1.41	0.01	−2.48	−0.33
	Hospital, oxygen	0.311	0.59	−0.83	1.44
	Hospital, NIV or ICU	1.22	0.07	−0.12	2.57
Hospital, oxygen	Home care	1.72	0.0003	−2.66	−0.78
	Hospital, no oxygen	−0.31	0.59	−1.44	0.82
	Hospital, NIV or ICU	0.91	0.15	−0.33	2.15
Hospital, NIV, or ICU	Home care	−2.63	0.0001	3.81	−1.44
	Hospital, no oxygen	−1.22	0.07	−2.57	0.12
	Hospital, oxygen	−0.91	0.15	−2.15	0.33

Legend: CI = confidence interval; NIV = non-invasive ventilation.

**Table 4 jcm-11-01774-t004:** Distribution of disease severity according to binary flow-mediated dilation (*p* < 0.0001 at chi-squared test).

FMD	Disease Severity					Total
	Home Care	Hospital Care				
		No Oxygen	Oxygen	NIV or ICU	Total	
>7.10%	44.5%	17.8%	24.9%	12.7%	55.4%	100%
≤7.10%	21.7%	20.8%	39.2%	18.3%	78.3%	100%

Legend: FMD = flow-mediated dilation; NIV = non-invasive ventilation.

**Table 5 jcm-11-01774-t005:** Multinomial regression analysis (reference category: not hospitalized).

Severity		*p*	OR	95% CI	
				Lower	Upper
Hospital, no oxygen	FMD ≤ 7.10%	0.005	2.39	1.29	4.42
Hospital, oxygen	FMD ≤ 7.10%	0.0001	3.22	1.88	5.51
Hospital, NIV, or ICU	FMD ≤ 7.10%	0.0009	2.96	1.55	5.65

Legend: CI = confidence interval; FMD = flow-mediated dilation; NIV = non-invasive ventilation; OR= odds ratio.

**Table 6 jcm-11-01774-t006:** Logistic regression analysis considering FMD as the dependent variable and disease severity as the independent variable.

		OR	95% CI	
	*p*		Lower	Upper
Disease severity	0.011	1.354	1.06	1.71
Age	0.0001	1.933	1.370	2.726
Sex	0.71	1.09	0.67	1.78
BMI	0.70	0.94	0.68	1.29
CRP	0.62	0.98	0.93	1.04
Arterial hypertension	0.48	0.82	0.47	1.42
T2DM	0.11	1.89	0.85	4.22
Smoking status	0.51	1.08	0.85	1.39
Constant	0.0001	0.06		

Legend: CI = confidence interval; BMI = body mass index; CRP = C-reactive protein; OR = odds ratio; T2DM= type 2 diabetes mellitus.

## Data Availability

The data presented in this study are available on request from the corresponding author. The data are not publicly available due to privacy reasons.

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
