# Peer review of "Impaired Endothelial Function in Convalescent Phase of COVID-19: A 3 Month Follow Up Observational Prospective Study"

_jcm, 2022, doi:10.3390/jcm11071774_

Round 1

Reviewer 1 Report

This observational study highlights the role of endothelial dysfunction in acute forms of covid 19 in convalescent patients examined for a 3-month follow-up. The measurement of flow mediated dilation (FMD) in the convalescence phase of the disease seems to show a persistence of the inflammatory state in the vessels that other inflammatory parameters do not detect. Furthermore, the FMD data observed in the different groups are directly correlated with pulmonary functions characterized as PaO2, P / F, FEV1 and 6MWT supporting the role of endothelial dysfunction in the development of the clinical manifestations of acute post COVID 19 syndrome.
An element that could add information to this work is the evaluation of the percentage of deaths in the different groups observed in relation to the values of FMD.

Author Response

Reviewer #1

Q1. This observational study highlights the role of endothelial dysfunction in acute forms of covid 19 in convalescent patients examined for a 3-month follow-up. The measurement of flow mediated dilation (FMD) in the convalescence phase of the disease seems to show a persistence of the inflammatory state in the vessels that other inflammatory parameters do not detect.

Furthermore, the FMD data observed in the different groups are directly correlated with pulmonary functions characterized as PaO2, P / F, FEV1 and 6MWT supporting the role of endothelial dysfunction in the development of the clinical manifestations of acute post COVID 19 syndrome.

An element that could add information to this work is the evaluation of the percentage of deaths in the different groups observed in relation to the values of FMD.

A1. We thank the reviewer for this interesting point. However, since the study was focused on post-acute COVID-19 recovery health check 3 months from the negative SARS-CoV-2 molecular test to a post-acute care service, established in April 21/2020, we have data available only on surviving patients; FMD was assessed in the first visit 3 months after infection, thus available only in subjects who survived the event. We have added a sentence to clarify this point (pag 2/13).

Reviewer 2 Report

This is an interesting report. Some issues need to be addressed:

  1. The term "Convalescent Phase" is used only in the title and the aims. Since this is not a widely accepted term for COVID-19, please define that in the introduction.
  2. Please clarify in the Abstract that the results have incorporated a multivariate analysis.
  3. What was the time period of recruitment?
  4. Have COVID-19 variants and/or vaccination influenced FMD values?
  5. Have the authors compared FMD values at diagnosis and 3 months later? This needs a paired test and would be useful.
  6. Please add sub-headings in the Results section, so that it becomes easier to read.
  7. Please take into account COVID-19 treatments. Do they influence FMD?
  8. Please also incorporate long COVID-19 in your analysis or at least discuss this.
  9. Please also discuss treatments that could influence endothelial dysfunction, either in the acute phase or long-term
  10. Although FMD is the gold standard method for endothelial dysfunction, it has many limitations that should be added in the limitations of the study. Furthermore, inter- and intra-rater variability should be reported.

Author Response

Reviewer #2
This is an interesting report.

We thank the Reviewer for this overall positive comment.

Some issues need to be addressed:

Q2. The term "Convalescent Phase" is used only in the title and the aims. Since this is not a widely accepted term for COVID-19, please define that in the introduction.

A2. We thank the reviewer for this improvement: in the revised version, we have added a sentence in the introduction section to better clarify this term (pag. 2/13).

Q3. Please clarify in the Abstract that the results have incorporated a multivariate analysis.

A3. As suggested by the reviewer, in the revised version we have added the term multivariate analysis in the abstract section (pag. 1/13).

Q4. What was the time period of recruitment?

A4. We thank the reviewer for this clarification. In the revised version, we have added a sentence to specify the timeframe of recruitment of the analyzed cohort (pag. 2/13).

Q5. Have COVID-19 variants and/or vaccination influenced FMD values?

A5. We thank the reviewer for this interesting point. This study refers to patients infected in late-2020 and first months of 2021, then we did only observe unvaccinated subjects infected by the original strain. Since we continued enrolling patients even after this wave, it will be interesting to point out this suggestion in a second paper analysing a larger cohort. We have added a sentence in the text, in the “future directions” subheading (pag. 11/13).

Q6. Have the authors compared FMD values at diagnosis and 3 months later? This needs a paired test and would be useful.

A6. We thank the reviewer for this question. We agree on the usefulness of this measurement, however we only assessed FMD 3 months after COVID-19. We clarified this point in the text, in the study limitations subheading (pag. 11/13).

Q7. Please add sub-headings in the Results section, so that it becomes easier to read.

A7. In the revised version, we have added sub-headings as requested (pag 4-5/13).

Q8. Please take into account COVID-19 treatments. Do they influence FMD?

A8. This question is interesting and should be addressed with larger studies: we did not consider this interesting point in the final analysis, since this datum was very heterogeneous in our database. In fact, we considered both hospitalized subjects and non-hospitalized patients and observed that non-hospitalized patients were substantially untreated, and that the sub-population of hospitalized patients was submitted to a very wide number of treatments, also ineffective, during the first time of pandemic. Thus, we were not able to weight the importance of each treatment on the outcome. We have added a sentence in the study limitations clarifying this point (pag. 11/13).

Q9. Please also incorporate long COVID-19 in your analysis or at least discuss this.

A9. This question is interesting. Unfortunately, we have no data about clinical long COVID-19 but we incorporate our hypothesis in the discussion (pag. 11/13).

Q10. Please also discuss treatments that could influence endothelial dysfunction, either in the acute phase or long-term

A10. This study was performed during the first time of pandemic without an effective therapy of COVID-19. We think that new treatment, especially anti-IL6 and anti-JAK, could influence endothelial disfunction in acute and convalescent phase. We have added a sentence in Future Directions (pag 11/13).

Q11. Although FMD is the gold standard method for endothelial dysfunction, it has many limitations that should be added in the limitations of the study. Furthermore, inter- and intra-rater variability should be reported.

A11. Thank you for the question.  We added in the study limitation that the clinical evaluation of endothelial function was carried out by calculating FMD, a technique that encloses different methodological approaches that can limit its validity and comparability, without the support of a dedicated software (pag. 11/13). Value of inter-observer variability was not available because of, as we specified in the methods of the manuscript, all evaluations were performed by a single skilled sonographer (pag 3/13). Furthermore, in the results section we have added the value of intra-rater correlation coefficient as requested (pag. 3-4/13).

Reviewer 3 Report

The manuscript is very well written and interesting and addresses an important topic – CVD risk complications determined by endothelial disfunction post COVID 19 infections, depending on the gravity of the disease.

The patient group is important and the statistical analysis considers all factors – age, sex, smoking, pathologies, etc. Results show a direct correlation between FMD and some variables of pulmonary function, suggesting a role of endothelial dysfunction in development of functional 311 and clinical manifestations of post-acute COVID-19 syndrome

Author Response

Reviewer #3

The manuscript is very well written and interesting and addresses an important topic – CVD risk complications determined by endothelial disfunction post COVID 19 infections, depending on the gravity of the disease. The patient group is important, and the statistical analysis considers all factors – age, sex, smoking, pathologies, etc. Results show a direct correlation between FMD and some variables of pulmonary function, suggesting a role of endothelial dysfunction in development of functional 311 and clinical manifestations of post-acute COVID-19 syndrome.

We thank the Reviewer for this overall positive comment.

Round 2

Reviewer 2 Report

The authors have adequately responded to reviewer's comments.

Author Response

This manuscript is a resubmission of an earlier submission. The following is a list of the peer review reports and author responses from that submission.